# Self-Prophetic Decoding to Unlock Visual Search in LVLMs

Zhendong He [* 1]    Qiyuan Dai [* 2]    Guanbin Li [1]    Liang Lin [1]    Sibei Yang [1]

## Abstract

Large Vision-Language Models (LVLMs) are rapidly evolving toward true multimodal reasoning, with visual search representing a concrete instantiation of the thinking-with-images paradigm. However, LVLM visual search faces two key challenges: incompatibility among intrinsic capabilities after post-training, and interference in long multi-step reasoning contexts. To address these, we identify two novel insights. First, self-regulation between pre- and post-training LVLMs leverages the intrinsic single-step capabilities of the pre-training model to mitigate capability deterioration and long-context interference. Second, probability-based prophetic sampling, replacing naive prompting, provides a probabilistic interface where the pre-training model acts as a prophet and the post-training model selectively accepts prophetic tokens under its output distribution, preserving coherent multi-step reasoning. Building on these insights, we introduce SeProD, a self-prophetic decoding framework that leverages intrinsic single-step capabilities to enable coherent multi-step reasoning in a training-free, plug-and-play manner. Experiments show that SeProD consistently improves multiple visual-search LVLMs across all 12 splits of 4 visual search benchmarks, as well as across general VQA benchmarks, without added computational overhead, thanks to its parallel prophetic acceptance mechanism.

## 1. Introduction

Large Vision-Language Models (LVLMs) are rapidly evolving from simple perception toward true multimodal reasoning. Recently, the thinking-with-images paradigm (Xu et al., 2025; Hu et al., 2024; Lai et al., 2025; Cheng et al., 2025; Fan et al., 2025; Liang et al., 2025) exemplifies the

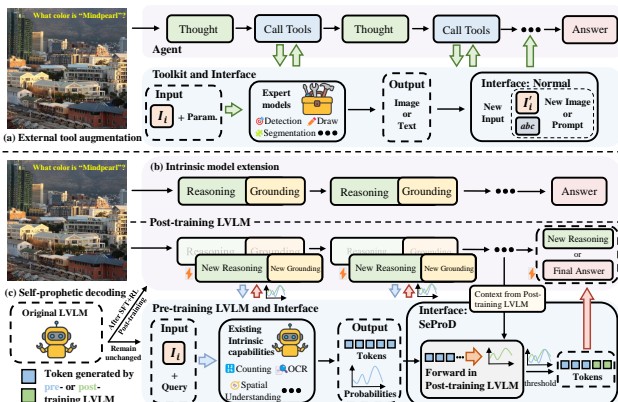

*Figure 1.* Overview of paradigms for enabling visual search in LVLMs. **(a) External tool augmentation.** LVLMs call visual tools and fuse tool outputs into subsequent reasoning, but the interface is rigid and fragments multi-step reasoning. **(b) Intrinsic model extensions.** LVLMs natively activate zoom-in and region grounding in a single forward pass, but visual-search post-training introduces incompatibilities among these intrinsic capabilities. **(c) Our SeProD** remains naturally compatible with LVLMs by operating at the pre-training level, while providing a flexible probabilistic interface to seamlessly integrate and coordinate these abilities during post-training visual search. Consequently, SeProD preserves intrinsic single-step capabilities like (a), while enabling coherent multi-step reasoning akin to (b).

generation and integration of visual evidence as intermediate reasoning steps, extending beyond text-only chain-of-thought (Chen et al., 2024b; Zheng et al., 2023; Mitra et al., 2024). In particular, visual search (Zhang et al., 2025b) constitutes a concrete and well-defined instantiation of thinking-with-images, in which correctly answering questions requires actively searching, grounding, and zooming in to verify specific regions within usually high-resolution images. This makes visual search a natural and tractable starting point for studying multimodal reasoning.

Recent efforts to endow LVLMs with visual search broadly fall into two categories: external tool augmentation and intrinsic model extension. The former employs function-calling or tool usage to provide LVLMs with tool-specific capabilities, the outputs of which are then integrated to LVLMs' subsequent reasoning (Li et al., 2025; Hu et al., 2024; Su et al., 2025; Wu et al., 2025). While enabling direct use of well-developed tools, it is constrained by rigid

---

*Equal contribution  [1]School of Computer Science and Engineering, Sun Yat-sen University [2]ShanghaiTech University. Correspondence to: Sibei Yang <yangsb3@mail.sysu.edu.cn>.

*Proceedings of the 43$^{rd}$ International Conference on Machine Learning*, Seoul, South Korea. PMLR 306, 2026. Copyright 2026 by the author(s).

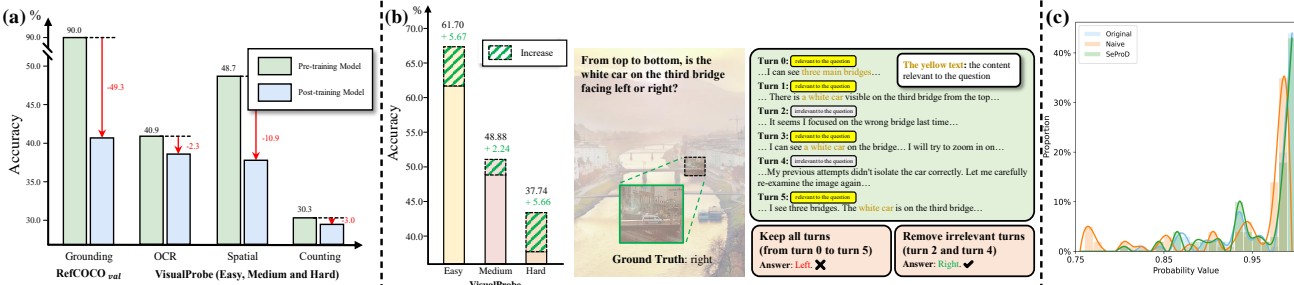

*Figure 2.* **(a) The degradation of intrinsic capabilities at a single step after visual-search post-training.** Performance drops on grounding, OCR, spatial understanding, and counting when evaluated at a specific reasoning turn. **(b) Interference accumulation in long multi-step trajectories.** Masking irrelevant context recovers correct predictions, indicating sensitivity to early-step errors. **(c) Distribution curves of the original visual-search LVLM, the naïve method, and our SeProD**, shown in blue, orange, and green, demonstrate that SeProD, by accepting only tokens aligned with the native distribution, preserves output consistency with the original model and thereby promotes coherent multi-step reasoning. Please refer to Appendix Sec. E for experimental details.

interfaces and the risk of errors from treating multi-step, context-dependent reasoning as multiple isolated, single-step task executions, as shown in Fig. 1(a). In contrast, intrinsic model extensions aim to directly activate LVLMs' native capabilities, such as zoom-in and region-of-interest grounding, during reasoning (Wang et al., 2025a; Zheng et al., 2025; Zhang et al., 2025b; Wang et al., 2025c; Lai et al., 2025). This allows closer integration between reasoning and intrinsic abilities, supporting cross-step knowledge sharing within a single forward pass, as shown in Fig. 1(b).

In this work, we focus on visual search, particularly complex scenarios that unfold over multiple interdependent reasoning steps, which motivates us to follow intrinsic model extensions. However, current intrinsic model extensions face two crucial challenges: *(1) Incompatibility among intrinsic capabilities after visual-search post-training.* Grounding, answering, and high-level reasoning are not jointly optimized in visual search. Limited instruction-following data restricts generalization, and reinforcement learning lacks supervision on intermediate steps. The optimization therefore favors task completion over maintaining robustness of individual steps that depend on different intrinsic abilities. Consequently, capabilities that originally functioned independently begin to interfere with each other, and some deteriorate or are even catastrophically forgotten. We assess single-step reasoning by selecting a specific step from the inference trajectory and comparing models (Bai et al., 2025b; Lai et al., 2025) with and without visual search post-training. As shown in Fig. 2(a), post-training reduces grounding, OCR recognition, spatial understanding, and counting performance by 49.3%, 2.3%, 10.9% and 3.0%, respectively. *(2) Interference in long multi-step reasoning contexts.* Long multi-step reasoning contexts, while essential for complex visual search, can in turn propagate interference, as mistaken intermediate predictions disrupting later reasoning and irrelevant information impairing the final answer, particularly when it mainly relies on visual

evidence from the last step. As shown in the left panel of Fig. 2(b), removing irrelevant content from the long reasoning trajectory yields improvements of 5.56%, 2.24%, and 5.66% on the three splits of VisualProbe-test (Lai et al., 2025), respectively. An example is provided in the right panel of Fig. 2(b).

To overcome these challenges, we introduce our *first key insight: the self-regulation of pre- and post-training LVLMs.* On one hand, since distinct intrinsic capabilities of pre-training LVLMs[1], e.g. grounding, zoom-in, and OCR recognition, already exist, they can be leveraged as internal guides to help the post-training LVLM integrate and coordinate these abilities, effectively addressing the challenge of incompatibility among intrinsic functions. On the other hand, post-training LVLMs can exploit cross-step shared context to dynamically steer the reasoning trajectory and focus of pre-training LVLMs for each single step. Simultaneously, feedback from these focused, single-step capabilities can mitigate the challenge of interference from long contexts while preserving reasoning coherence.

Despite these advantages, a new technical challenge arises: *(3) How to design an effective interface to enable interaction between pre- and post-training LVLMs?* A naïve approach is to feed textual prompts from the pre-training model as additional input prefixes or to overwrite the post-training model's current output. However, this reduces the interaction to a tool-like interface and introduces two issues. First, the prompts often fail to directly steer the post-training LVLM's outputs, limiting their effect on subsequent reasoning. Second, even when they do influence the outputs, their single-step nature can disrupt contextual coherence, destabilizing multi-step reasoning, as shown in Appendix Fig. 8. To address this, we propose our *second*

---

[1]In this paper, "pre-training" refers to the stage before visual search post-training (corresponding to base LVLMs like Qwen-2.5-VL), not the conventional LVLM pre-training.

*key insight: probability-based prophetic sampling as the interface*, as shown in Fig. 1(c). In prophetic sampling, the pre-training LVLM acts as a "prophet," and the post-training LVLM accepts tokens generated by this prophet if their probability under the adjusted distribution derived from post-training itself and the prophet exceeds a predefined threshold; otherwise, tokens are sampled from the post-training model. This mechanism allows the post-training LVLM to treat pre-training outputs directly as generated tokens rather than inputs, propagating accepted prefixes through subsequent decoding. Meanwhile, by only accepting tokens that align with its native distribution, the post-training model preserves its output consistency and maintains coherent multi-step reasoning, as evidenced by the blue and green distribution curves in Fig. 2(c).

To realize these insights, we propose a self-prophetic decoding (SeProD) framework for visual search, harnessing LVLMs' intrinsic capabilities to improve multi-step reasoning in a training-free, plug-and-play manner. Specifically, the post-training LVLM serves as the search model, dynamically steering the reasoning trajectory and executing visual search in a single forward pass. At each reasoning step, the current input and partial output are fed to the pre-training LVLM, acting as a prophet, which generates single-step, capability-specific prophetic prefixes. Then, the search model evaluates all prophetic prefixes in parallel, selectively accepting those that align with its output distribution, ensuring controlled and coherent propagation. Such interaction between the search and prophet models iterates over the reasoning process. Notably, prefix evaluation is fully parallelizable, and the prophet model can be a smaller counterpart, incurring minimal overhead and potentially improving efficiency. Moreover, because this sampling is imposed solely at inference and is compatible with any intrinsic-extended LVLM for visual search, SeProD is inherently training-free and plug-and-play.

In summary, our contributions are multi-faceted:

• We identify a new insight that self-regulation between pre- and post-training LVLMs leverages intrinsic single-step capabilities to mitigate deterioration and long-context interference caused by visual search training.
• We are the first to propose a probability-based prophetic interface that treats pre-training outputs as prophetic predictions, accepting them approximately under the post-training distribution to preserve coherence reasoning.
• We introduce a self-prophetic decoding framework (SeProD) that enables the activation of intrinsic single-step capabilities while supporting multi-step coherent reasoning in a training-free, plug-and-play manner.
• SeProD consistently improves multiple visual-search LVLMs, achieving state-of-the-art results on all 12 splits across 4 visual search benchmarks. It also yields gains on

several general VQA benchmarks, while introducing no additional computational overhead.

## 2. Related Work

**Large Vision-Language Models** (LVLMs) constitute a central line of research in multimodal learning, aiming to align visual representations with large language models for unified understanding and reasoning. Early works, such as BLIP-2 (Li et al., 2023) and LLaVA (Liu et al., 2023), established a paradigm that combines pre-trained vision encoders with large language models, enabling effective alignment between visual and linguistic modalities. To support more flexible image inputs and finer-grained visual perception, recent studies adopt the AnyRes(Li et al., 2024) strategy to overcome resolution constraints in input images. Building upon these advances, a series of strong LVLMs have emerged, including LLaVA (Li et al., 2024; Xu et al., 2024), Qwen-VL (Wang et al., 2024a; Bai et al., 2025b;a), and InternVL (Chen et al., 2024a; Gao et al., 2024).

**Multimodal Reasoning.** To enable sophisticated reasoning, existing methods generally follow two main categories. The first category transforms visual inputs into structured or symbolic representations (Yang et al., 2019; Zheng et al., 2023; Shao et al., 2024; Chen et al., 2024b), such as object relationship graph, captions and scene graphs, and then performing symbolic or language-based reasoning over these intermediate representations. The second category follows the thinking-with-images paradigm, in which LVLMs perform iterative, multi-step reasoning by dynamically interacting with visual inputs at inference time. Within this paradigm, some approaches (Surís et al., 2023; Hu et al., 2024; Su et al., 2025) rely on external tools, such as visual experts or code execution modules, to progressively refine visual evidence during reasoning. In contrast, other methods (Shen et al., 2025a; Fan et al., 2025; Zhang et al., 2025b; Wang et al., 2025a; Zheng et al., 2025; Wang et al., 2025c; Lai et al., 2025) stimulate the model's intrinsic capabilities, such as grounding, enabling it to capture fine-grained visual cues and support complex reasoning patterns.

**Visual Search** is a challenging yet essential multimodal capability within the thinking-with-images paradigm. It requires models to answer questions over high-resolution images by actively identifying and localizing regions relevant to fine-grained visual details. Some existing works (Wu & Xie, 2023; Liu et al., 2025; Li et al., 2025; Shen et al., 2025b; Zhang et al., 2025a) augment LVLMs with external modules or tools to perform visual search. In contrast, other approaches enable LVLMs to exploit their intrinsic grounding capability for region localization and zoom-in operations through end-to-end training (Zhang et al., 2025b; Wang et al., 2025c;a; Zheng et al., 2025; Lai et al., 2025). For example, ZoomEye (Shen et al., 2025b) employs a tree

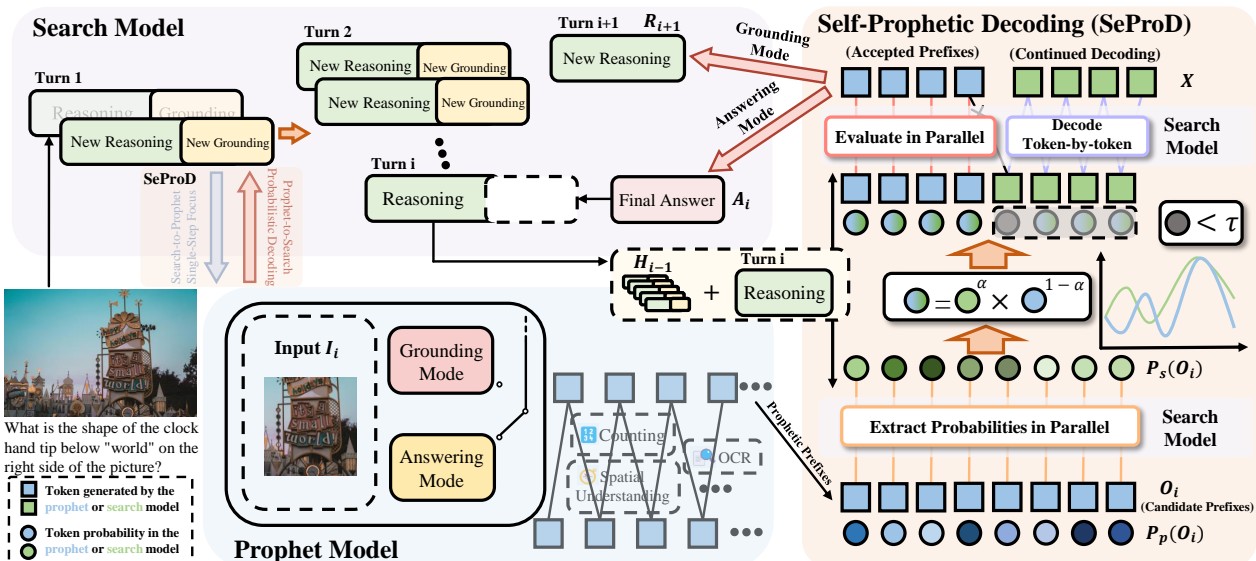

*Figure 3.* **The overall framework of SeProD. (1) Pair of search model and prophet model.** The post-training LVLM serves as the search model, responsible for steering the global multi-turn reasoning process. Its pre-training counterpart acts as the prophet model, exploiting native intrinsic capabilities to produce single-step prophetic prefixes. The two models are coupled through bidirectional signals: search-to-prophet specifies task-relevant focus at each turn, while prophet-to-search provides guidance for subsequent reasoning. **(2) Self-prophetic decoding** incorporates prophetic prefixes into the search model's decoding process. Candidate prefixes are evaluated in parallel and accepted only when they fall under the adjusted distributions of both models. Once accepted, they are absorbed as native tokens of the search model and propagated through subsequent token-by-token decoding, preserving the original generation dynamics.

search algorithm to simulate zoom-in and zoom-out operations, allowing the model to inspect image details and identify key information. DeepEyes (Zheng et al., 2025) and Mini-o3 (Lai et al., 2025) further leverage reinforcement learning to enable models to select zoom-in regions based on their intrinsic grounding capability, thereby acquiring task-relevant visual details. However, they are constrained by rigid interfaces, incompatibilities among intrinsic abilities, or interference from long contexts. SeProD addresses these limitations by enabling self-regulation and adopting probability-based sampling as the interface.

## 3. Method

The overall framework of self-prophetic decoding (SeProD) is illustrated in Fig. 3. SeProD enhances post-training LVLMs for visual search by integrating a pre-training counterpart that provides complementary intrinsic capabilities and single-step regulation via distribution-approximately prophetic sampling. First, we formalize the post-training LVLM as the search model and its pre-training counterpart as the prophet model (see Sec. 3.1). The search model dynamically steers the overall reasoning trajectory, while the prophet generates single-step, capability-specific prophetic prefixes, with the two models interacting iteratively. Second, in the search-to-prophet interaction, the search model guides the prophet's focus at each reasoning step, allowing it to generate prophetic prefixes for tasks focusing on dis-

tinct abilities such as grounding and counting (see Sec. 3.2). Third, in the prophet-to-search interaction, the prophetic prefixes are selectively incorporated into the search model via probability-based prophetic sampling, directly shaping its outputs while preserving the native distribution, thereby maintaining coherent multi-step reasoning (see Sec. 3.3).

### 3.1. Definition of Search and Prophet Models

We define a pair of search and prophet models in SeProD. The search model refers to any LVLM that has undergone post-training for visual search, e.g., Mini-o3 (Lai et al., 2025), DeepEyes (Zheng et al., 2025), and Pixel Reasoner (Wang et al., 2025a), or other intrinsic-extended LVLMs. Its prophet model is the corresponding pre-training counterpart, i.e., the same base LVLMs before visual search post-training. This pairing captures a complementary asymmetry: post-training equips the search model with long-horizon reasoning and visual-search priors, while the prophet preserves strong single-step capabilities.

**Search Model** continuously adjusts and steers the global reasoning trajectory. Specifically, given an image $I$, typically at high resolution, and a query $Q$, the search model performs multi-turn reasoning and generation in a single forward pass. At each reasoning turn $i$, it generates the output $(I_i, C_i)$ conditioned on the entire history $H_{i-1}$:

$$H_{i-1} = \{(I, Q), (I_1, C_1), \dots, (I_{i-1}, C_{i-1})\}. \quad (1)$$

Here, $I_{i-1}$ denotes the cropped then zoomed-in image produced from the previous step $i-1$, and $C_{i-1}$ is the corresponding textual output. The output $C_i$ is generated autoregressively as follows,

$$p_s(C_i \mid H_{i-1}) = \prod_{j=1}^{L_c} p_s(c_{i,j} \mid H_{i-1}, c_{i,<j}), \quad (2)$$

where $c_{i,j}$ and $c_{i,<j}$ denote the $j$-th token and its preceding tokens in $C_i$, respectively, and $L_c$ is the token length of $C_i$. For simplicity, and without loss of generality, we assume that all $C_i$ share the same token length. In particular, each $C_i$ and its corresponding $I_i$ are generated in two modes:

• Grounding mode: $C_i = (R_i, G_i)$, where $R_i$ are reasoning tokens and $G_i$ is a grounding prediction, specifying a reference to a previously observed image (from the original image up to turn $i-1$) and a region of interest coordinate within it. The output image $I_i$ is obtained by cropping the referenced image at this region and zooming in.
• Answering mode: $C_i = (R_i, A_i)$, where $A_i$ contains the final answer. In this mode, the current turn terminates, and no further output image is generated. For notational convenience, we set image output $I_i = I_{i-1}$ in this case.

Notably, while each reasoning step yields only two output modes, these merely represent the form of the output; in reality, each step may engage distinct capabilities, such as grounding, OCR recognition, counting, and more.

**Prophet Model**, fortunately defined as the search model's base LVLM prior to visual-search post-training, perfectly leverages its native intrinsic abilities to serve as a complementary aid to the search model. Moreover, we select this counterpart because its general output distribution closely aligns with the search model, enhancing the acceptance rate in the self-prophetic decoding of Sec. 3.3. Specifically, given an image $I$ and a query $Q$, the prophet model generates the output $O$ directly in a single turn as follows,

$$p_p(O \mid I, Q) = \prod_{i=1}^{L_d} p_p(o_i \mid I, Q, o_{<i}), \quad (3)$$

where $o_i$ denotes the token at position $i$, $o_{<i}$ denotes all preceding tokens, and $L_d$ is total token sequence length.

## 3.2. Search-to-Prophet Single-Step Focus

In the search-to-prophet interaction, while the search model encompasses a multi-step reasoning trajectory, it provides step-specific guidance, directing the prophet's focus at each turn and enabling the generation of prophetic prefixes tailored to distinct task-specific abilities.

Specifically, at the $i$-th reasoning turn of the search model, the prophet model takes the target's output image $I_i$ as

input. The textual output $C_i$ is omitted to avoid biasing the draft toward the target's reasoning trajectory, preserving its independent, single-step capabilities. To accommodate the two modes of $C_i$—grounding and answering—we introduce two distinct queries: a grounding query, which verifies the validity of the current cropped image and steers subsequent reasoning, and an answering query, which produces answer drafts. The prophet model's prediction $O_i$ at this turn is then formulated as follows,

$$p_p(O_i \mid I_i, Q^p) = \prod_{j=1}^{L_d} p_p(o_{i,j} \mid I_i, Q^p, o_{i,<j}),$$

$$Q^p = \begin{cases} Q^g, & \text{if } C_i = (R_i, G), \\ Q, & \text{otherwise.} \end{cases} \quad (4)$$

Here, $Q^p$ denotes the prophet model's query. When the search model's output $C_i$ is in grounding mode, $Q^p$ is set to the grounding verification query $Q^g$; otherwise, it defaults to the original query $Q$ for question answering. The prophet model consistently uses the current turn's output image $I_i$ as its visual input. Next, we provide a detailed specification for these two modes.

**Grounding Verification for Next-Step Steering** first determines whether the current image crop contains the relevant regions for answering the query and, based on this feedback, generates prophetic prefixes for the search model's next turn. Specifically, it evaluates the presence of regions-of-interest in the current image $I_i$, yielding a binary response, true or false. If true, more detailed information about the region is included in the prophet model's output $O_i$, serving as prophetic prefixes for generating the reasoning token $R_{i+1}$ in the next turn $i+1$ (see Sec. 3.3 Eq. (5)). If false, the search model is prompted to relocate the potential region.

**Answer Drafting for Final Correction** directly produces the prophet model's answer $O_i$, which is used as prophetic prefixes for the search model's final answer $A_i$ via prophetic-to-search sampling. In this mode, $A_i$ is not generated beforehand; it is obtained on-the-fly through prophetic sampling for efficiency.

In summary, the prophet model's outputs $O_i$ (with their probabilities $p_p(O_i \mid I_i, Q^p)$) serve distinct roles for the search model: supplying prophetic prefixes for next-turn reasoning tokens $R_{i+1}$ in grounding mode, and prophetic prefixes for the current-turn answer $A_i$ in answering mode.

## 3.3. Prophet-to-Search Probabilistic Decoding

In the prophet-to-search interaction, the prophet model's output at the current turn, derived from its intrinsic single-step capabilities during the search-to-prophet phase, in turn feeds back to guide the search model's subsequent reasoning, grounding and answering. As discussed in Sec. 1, a straight-

forward interface that directly feeds the prophet model's output as an additional input to the search model either fails to meaningfully influence its generation or overly constrains, disrupting reasoning coherence. We thus propose a novel probability-based self-prophetic decoding to address these issues. Inspired by LLM speculative decoding (Leviathan et al., 2023) for inference acceleration, we propose a novel probability-based self-prophetic decoding, introduced for the first time to LVLM reasoning and visual search.

Specifically, at the $i$-th reasoning turn, the prophet model's output $O_i$ (defined at Eq. (4)) serves as prophetic prefixes for the search model to generate $R_{i+1}$ or $A_i$. For notational simplicity, we denote the resulting output after self-prophetic decoding as $X$, with $X \in \{R_{i+1}, A_i\}$. Generation of $X$ proceeds in two steps. First, the search model evaluates all tokens in $O_i$ and their probabilities $p_p(O_i \mid I_i, Q^p)$ in parallel, selectively accepting those approximately consistent with both the search and prophet models' distributions. This evaluation accelerates the search model's inference, as the tokens in $O_i$ are pre-collected and can be evaluated in parallel, rather than being generated sequentially. Second, for the first token that is not accepted, an additional token is sampled from the search model's distribution, and generation continues iteratively until $X$ is complete. The sampling of the $j$-th token of $X$ during decoding is formulated as follows:

$$
x_j \sim \begin{cases} p_p(x_j \mid I_i, Q^p, x_{<j}), & \text{if } j < \min\{j \mid s_j < \tau\}, \\ p_s(x_j \mid H_i, x_{<j}), & \text{otherwise,} \end{cases}
$$

where

$$
s_j = p_s(o_{i,j} \mid H_i, o_{i,<j})^\alpha \, p_p(o_{i,j} \mid I_i, Q^p, o_{i,<j})^{1-\alpha} \tag{5}
$$

Here, $s_j$ measures the consistency of the $j$-th token $o_{i,j}$ in $O_i$ with both the search and prophet model distributions, specifically $p_s(o_{i,j} \mid H_i, o_{i,<j})$ and $p_p(o_{i,j} \mid I_i, Q^p, o_{i,<j})$, with higher values indicating stronger agreement. And $\alpha$ is a balancing factor, initialized to $0.5$ and automatically adjusted based on the normalized relative rank of token $o_{i,j}$ in the search model's logits. A higher rank leads to a larger $\alpha$, promoting stronger adherence to the search model's native distribution. $\tau$ serves as a consistency threshold hyperparameter that determines the acceptance boundary for draft tokens $O_i$. When the score $s_j \geq \tau$, the search model accepts the prophet model's output $o_{i,j}$. Conversely, $\min\{j \mid s_j < \tau\}$ identifies the index of the first rejected token, from which the search model resumes generation solely based on its native distribution $p_s(x_j \mid H_i, x_{<j})$.

Notably, the proposed probability-based self-prophetic decoding allows the search model to accept the prophet model's outputs as valid prefixes conforming to its own distribution, thereby fully leveraging single-round capabilities while maintaining coherence across multi-turn reasoning. Using the proposed decoding, the search model generates

the output $X$. As before, in grounding mode, $X$ corresponds to the next-turn reasoning tokens $R_{i+1}$, while in answering mode it represents the current-turn answer $A_i$. The search-to-prophet and prophet-to-search interactions then alternate iteratively until the final answer is produced.

## 4. Experiment

**Benchmarks.** We evaluate SeProD using two categories of benchmarks. (1) *Visual Search.* We use V\* Bench (Wu & Xie, 2023), HR-Bench (Wang et al., 2024b), and Visual-Probe test (Lai et al., 2025). These benchmarks focus on high-resolution visual reasoning and require precise localization of small visual entities. They test whether an LVLM can extract task-relevant information from complex visual scenes and handle fine-grained spatial distinctions. (2) *General VQA.* We use MME-RealWorld (Zhang et al., 2025c) as the comprehensive general VQA benchmark. In addition, we include ScienceQA (Lu et al., 2022), OCRBench (Liu et al., 2024), and CVBench (Tong et al., 2024) as benchmarks that aim to reflect intrinsic LVLM capabilities such as OCR recognition, spatial understanding, and counting. Details of the benchmarks are provided in Appendix Sec. D.

**Baselines.** To verify the transferability and robustness of our framework, we select three representative post-training LVLMs, i.e., Pixel Reasoner (Wang et al., 2025a), Deep-Eyes (Zheng et al., 2025), and Mini-o3 (Lai et al., 2025), as baselines. These models vary substantially in both architecture and post-training strategies. By applying our approach to each of them without modification, we demonstrate that it can be seamlessly integrated into diverse model families and reasoning pipelines.

**Implementation Details.** We use Qwen2.5-VL-3B (Bai et al., 2025b) as the default prophet model for efficiency, unless explicitly stated otherwise. We use the official pipelines of each baseline and apply SeProD during inference. The consistency threshold $\tau$ is fixed to $0.3$ and set $\alpha = 0.5 - r$, where $r$ denotes the normalized rank of the corresponding token in the search model's output logits. We adopt either the standard accuracy or the averaged accuracy. Specifically, VisualProbe test is evaluated using avg@32 for all baselines, while for V\* Bench and HR-Bench, we use standard accuracy for Pixel Reasoner and DeepEyes. Following the evaluation metrics of Mini-o3, avg@32 and avg@8 are used for the two benchmark respectively for Mini-o3.

### 4.1. Comparison with State-of-the-Art Methods

Tab. 1 presents a comparative analysis of SeProD against state-of-the-art approaches on all high-resolution benchmarks. We compare the performance of the method both with and without external tools. Across all the benchmarks, our approach consistently yields notable gains over the

*Table 1.* **Comparison with state-of-the-art methods on high-resolution benchmarks.** Performance of SeProD evaluated on VisualProbe-test, V* Bench, and HR-Bench across multiple baseline LVLMs. **SeProD consistently improves over the original models on all benchmarks and subsets, including both open-ended reasoning and multiple-choice settings.** The gains are particularly pronounced on challenging scenarios with long reasoning trajectories and strong spatial or cross-instance perception requirements. Results marked with † are reproduced using the official code. "7B" denotes the prophet model size; otherwise, it defaults to 3B.

| Method | VisualProbe | | | V* Bench | | | HR-Bench 4K | | | HR-Bench 8K | | |
|---|---|---|---|---|---|---|---|---|---|---|---|---|
| | Hard | Medium | Easy | Attr. | Spatial | Overall | FSP | FCP | Overall | FSP | FCP | Overall |
| *Methods with external tool* | | | | | | | | | | | | |
| SEAL (Wu & Xie, 2023) | - | - | - | 74.8 | 76.3 | 75.4 | - | - | - | - | - | - |
| ZoomEyes (Shen et al., 2025b) | - | - | - | 93.9 | 85.5 | 90.6 | 84.3 | 55.0 | 69.6 | 88.5 | 50.0 | 69.3 |
| RAP (Wang et al., 2025b) | - | - | - | 90.4 | 96.1 | 91.1 | 73.8 | 40.5 | 57.1 | 72.3 | 35.3 | 53.8 |
| DyFo (Li et al., 2025) | - | - | - | 80.0 | 82.9 | 81.2 | - | - | - | - | - | - |
| FOCUS (Zhong et al., 2025) | - | - | - | - | - | 90.6 | - | - | 79.3 | - | - | 76.3 |
| *Methods without external tool* | | | | | | | | | | | | |
| Chain-of-Focus (Zhang et al., 2025b) | - | - | - | - | - | 88.0 | - | - | - | - | - | - |
| Simple-o3 (Wang et al., 2025c) | - | - | - | - | - | 90.4 | - | - | 76.2 | - | - | - |
| Pixel Reasoner† (Wang et al., 2025a) | 28.7 | 29.0 | 58.7 | 88.7 | 84.2 | 86.9 | 84.8 | 60.5 | 72.6 | 76.3 | 52.3 | 64.3 |
| Pixel Reasoner + Ours | 30.2 | 30.4 | 61.7 | 90.4 | 85.5 | 88.5 | 86.0 | 61.3 | 73.6 | 77.5 | 52.8 | 65.1 |
| Δ | +1.5 | +1.4 | +3.0 | +1.7 | +1.3 | +1.6 | +1.2 | +0.8 | +1.0 | +1.2 | +0.5 | +0.8 |
| DeepEyes† (Zheng et al., 2025) | 38.4 | 30.5 | 61.2 | 90.4 | 86.8 | 89.0 | 90.8 | 55.3 | 73.0 | 85.3 | 54.5 | 69.9 |
| DeepEyes + Ours | 41.9 | 32.3 | 64.7 | 91.3 | **90.8** | **91.1** | 91.8 | 55.8 | 73.8 | 85.8 | 58.0 | 71.9 |
| Δ | +3.5 | +1.8 | +3.5 | +0.9 | +4.0 | +2.1 | +1.0 | +0.5 | +0.8 | +0.5 | +3.5 | +2.0 |
| Mini-o3† (Lai et al., 2025) | 47.2 | 49.0 | 66.7 | 88.8 | 82.4 | 86.3 | 90.4 | 64.0 | 77.2 | 88.9 | 57.0 | 73.0 |
| Mini-o3 + Ours | 50.5 | 51.5 | 69.6 | 90.8 | 86.8 | 89.2 | 91.5 | **65.2** | 78.3 | 90.2 | 59.7 | 75.0 |
| Δ | +3.3 | +2.5 | +2.9 | +2.0 | +4.4 | +2.9 | +1.1 | +1.2 | +1.1 | +1.3 | +2.7 | +2.0 |
| Mini-o3 + Ours (7B) | **51.5** | **52.2** | **71.3** | **92.0** | 88.0 | 90.4 | **92.0** | **65.2** | **78.6** | **90.5** | 59.9 | **75.2** |
| Δ | +4.3 | +3.2 | +4.6 | +3.2 | +5.6 | +4.1 | +1.6 | +1.2 | +1.4 | +1.6 | +2.9 | +2.2 |

baselines, underscoring its effectiveness in enhancing post-training LVLMs via probability-guided prophetic sampling, which enables a more synergistic interaction between pre- and post-training models.

**Results on VisualProbe test.** SeProD substantially improves the performance of all baselines on VisualProbe test, which is designed to encourage reflective, trial-and-error reasoning (Lai et al., 2025). Notably, on the easy subset, SeProD achieves improvements of 3.0%, 3.5%, and 2.9% over the three baselines, respectively. On the more challenging hard subset, SeProD yields gains of 3.5% and 3.3% for DeepEyes and Mini-o3, respectively. Due to the presence of numerous distracting elements and open-ended answers, the VisualProbe is substantially more challenging than V*Bench and HR-Bench. The larger performance gains achieved by SeProD on VisualProbe demonstrate its effectiveness in enhancing the search model's ability to locate and perceive fine-grained targets in complex visual search scenarios.

**Results on V* Bench and HR-Bench.** Performance gains are also observed on multiple-choice benchmarks such as V* Bench and HR-Bench (4K and 8K), as well as on each of their respective subsets. Our approach improves accuracy across all baselines, confirming its robustness beyond

open-ended reasoning tasks. Notably, SeProD enables both DeepEyes and Mini-o3 to perform particularly well on tasks involving multiple instances. On the Spatial Understanding subset of V*Bench, it improves performance by 4.0% and 4.4%, respectively, while on the FCP subset of HR-Bench 8K, the corresponding gains are 3.5% and 2.7%. These gains suggest that SeProD effectively leverages the intrinsic spatial reasoning capability of the prophet model to guide inference, thereby enabling the search model to better identify and reason about complex spatial relations. Similarly, the consistent performance improvements observed for Deep-Eyes and Mini-o3 on the FCP subset of the HR Bench-8K benchmark indicate that the prophet model produces more precise outcomes for cross-instance perception. This guidance helps the search model refocus on identifying relationships among multiple instances, leading to more reliable predictions in challenging high-resolution scenarios.

**Results on General VQA benchmarks.** Using Mini-o3 as the search model, Qwen2.5-VL-3B as the prophet model, we compare the performance of Qwen2.5-VL-7B, Deep-Eyes, Mini-o3, and SeProD on a diverse set of general VQA benchmarks. As shown in Tab. 2, SeProD not only improves the search model's performance on visual search tasks but also achieves better performance on the general VQA bench-

*Table 2.* **Results on general VQA benchmarks.** Performance comparison of baselines and SeProD on general VQA benchmarks. SeProD consistently improves the visual search LVLM and achieves superior performance.

| Method | MME-RW | ScienceQA | OCRBench | CVBench |
|---|---|---|---|---|
| Qwen2.5-VL | 57.3 | 69.4 | 81.5 | 73.9 |
| DeepEyes | 64.0 | - | - | - |
| Mini-o3 | 65.5 | 84.5 | 83.8 | 74.4 |
| Ours | **67.7** | **85.4** | **85.3** | **78.4** |

marks. The performance gains on MME-RealWorld and CVBench are particularly notable. By integrating SeProD, the intrinsic capabilities of the search model are further activated and reinforced, resulting in a more substantial improvement on these benchmarks.

*Table 3.* **Computational Overhead of SeProD.** Acceptance rate of prophetic-prefix tokens under SeProD and the resulting inference speedup relative to the baseline. Results indicate high acceptance rates and slight speedups.

| Metric | VisualProbe | | |
|---|---|---|---|
| | **Hard** | **Medium** | **Easy** |
| Acceptance rate | 74.6% | 74.2% | 80.7% |
| Speedup | **1.06x** | **1.03x** | **1.07x** |

**Computational Overhead.** We evaluate the computational cost incurred by SeProD across the entire inference pipeline. Specifically, we conduct an empirical analysis using Mini-o3 on the VisualProbe test dataset, and report both the proportion of prophetic-prefix tokens accepted by the search model and the speedup ratio. In terms of implementation, we use NVIDIA H20 GPUs to run the search model and the prophet model. For each subset of VisualProbe test, inference is conducted 4 times with a batch size of 4. The results are reported in Tab. 3. Across all subsets of the VisualProbe test, SeProD achieves high acceptance rates without introducing additional computational overhead, owing to the parallel evaluation of prefixes in the search model.

### 4.2. Ablation Study

In this section, we employ Mini-o3 as the base model to conduct ablation experiments, aiming to isolate the effects of individual components in SeProD and examine how varying the scale of the prophet model impacts overall performance.

**Component Effectiveness.** As shown in Tab. 4, (1) the naïve approach performs worse than the search model itself most of the time, indicating that simply feeding the pre-training model's output as textual prompts or overriding the post-training model's current output does not lead to meaningful improvements; instead, it may interfere with or even degrade the model's existing capabilities. (2) We evaluate

*Table 4.* **Ablation results.** Effectiveness of individual components in SeProD (left table), as well as the impact of different thresholds (right table). The results show that each component contributes to the performance gains. SeProD achieves the best results, and SeProD is robust to the choice of threshold. "S w/o A" and "S w/o G" denote SeProD without Answer Drafting and Grounding Verification, respectively.

| Method | VisualProbe | | | $\tau$ | VisualProbe | | |
|---|---|---|---|---|---|---|---|
| | Hard | Medium | Easy | | Hard | Medium | Easy |
| naïve | 46.2 | 47.4 | 67.0 | 0.2 | 50.0 | 51.0 | 69.0 |
| S w/o A | 49.5 | 50.1 | 68.1 | 0.25 | 50.3 | 50.9 | 69.3 |
| S w/o G | 49.9 | 50.4 | 68.7 | 0.3 | **50.5** | **51.5** | **69.6** |
| Ours | **50.5** | **51.5** | **69.6** | 0.35 | 50.0 | 51.2 | 69.2 |

Grounding Verification and Answer Draft independently. Both modules provide complementary strengths. Grounding Verification improves spatial grounding precision and facilitates deeper exploration of task-relevant regions, while the Answer module enhances the semantic plausibility of candidate responses. Their integration yields the largest performance gain, highlighting the importance of jointly verifying the model's grounding and supplying semantically plausible candidate answers. (3) In addition, we investigate the effect of different threshold values $\tau$ on SeProD. The results show that a wide range of reasonable thresholds can consistently yield performance gains. This demonstrates that our approach is robust to hyperparameter choices and that the improvements arise from its intrinsic effectiveness rather than from hyperparameter tuning.

*Table 5.* **Effect of alpha configuration.** Fixed $\alpha$ improves the baseline model, but underperforms dynamic $\alpha$ due to its inability to adapt to token-level uncertainty and confidence.

| $\alpha$ | VisualProbe | | |
|---|---|---|---|
| | **Hard** | **Medium** | **Easy** |
| 0.3 | 49.3 | 49.8 | 68.6 |
| 0.4 | 48.7 | 50.1 | 67.9 |
| 0.5 | 48.6 | 49.6 | 67.6 |
| 0.6 | 49.1 | 50.0 | 68.8 |
| 0.7 | 48.7 | 49.1 | 69.0 |
| 0.5 - $r$ | **50.5** | **51.5** | **69.6** |

**Alpha configuration.** We compare a dynamically adjusted $\alpha$ with a fixed $\alpha$. As shown in Tab. 5, a fixed $\alpha$ improves the performance of the baseline model, thanks to the probability-based prophetic sampling. However, its gains are smaller than those of a dynamic $\alpha$, because a fixed $\alpha$ cannot adaptively adjust to the uncertainty of individual tokens, nor can it leverage the search model's confidence for each token to dynamically allocate weights.

**Prophet Model Scale.** Tab. 1 further evaluates a 7B prophet model for comparison with the default 3B setting. The 7B

prophet yields additional performance gains, particularly on the more challenging VisualProbe benchmark. Beyond the inherent advantage of larger model capacity, a potential contributing factor is that the search model is also initialized from the same 7B model, resulting in closer output distributions between the search model and the prophet model.

## 5. Conclusion

In this paper, we introduce SeProD, a training-free and plug-and-play framework that aligns pre- and post-training LVLMs to address capability degradation and long-context interference in visual search. By leveraging a probability-based prophetic interface, SeProD activates precise single-step abilities while preserving contextual coherence. Experiments show that our approach consistently improves multiple intrinsic-extended LVLMs and achieves state-of-the-art performance across all visual search benchmarks, and further enhances performance on general VQA benchmarks, without introducing additional computational overhead.

## Impact Statement

This work aims to advance the field of machine learning by improving the coherence of multi-step multimodal reasoning in large vision-language models. The proposed framework focuses on inference mechanisms for coordinating intrinsic model capabilities, without introducing new data sources, training procedures, or application-specific objectives. We do not foresee direct negative societal consequences arising uniquely from this work. Potential risks and ethical considerations are therefore aligned with those already well established for Large Vision-Language Models such as issues related to bias, misuse, or over-reliance on automated reasoning systems.

## Acknowledgment

This work is supported by the National Natural Science Foundation of China under Grant No.62576365 and in part by the National Natural Science Foundation of China under Grant No. 62322608.

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

In this appendix, we provide comprehensive information, including examples and failure cases of SeProD, failure of naïve textual interfaces, details on the benchmarks used in this paper, and implementation details for analysis in the introduction.

- Sec. A - Examples of SeProD
- Sec. B - Failure Cases of SeProD
- Sec. C - Failure of Naïve Textual Interfaces
- Sec. D - More Details on the Benchmarks Used in This Paper
- Sec. E - Implementation Details for Analysis in the Introduction

# A. Examples of SeProD

In this section, we present some examples illustrating how SeProD leverages the prophet model to help the search model obtain better results. Specifically, Fig. 4 shows the case where the prophet model's output serves as prophetic prefixes to generate the reasoning token in the next turn. Fig. 5 demonstrates how the search model is guided to perform a further zoom-in to obtain a more accurate answer.

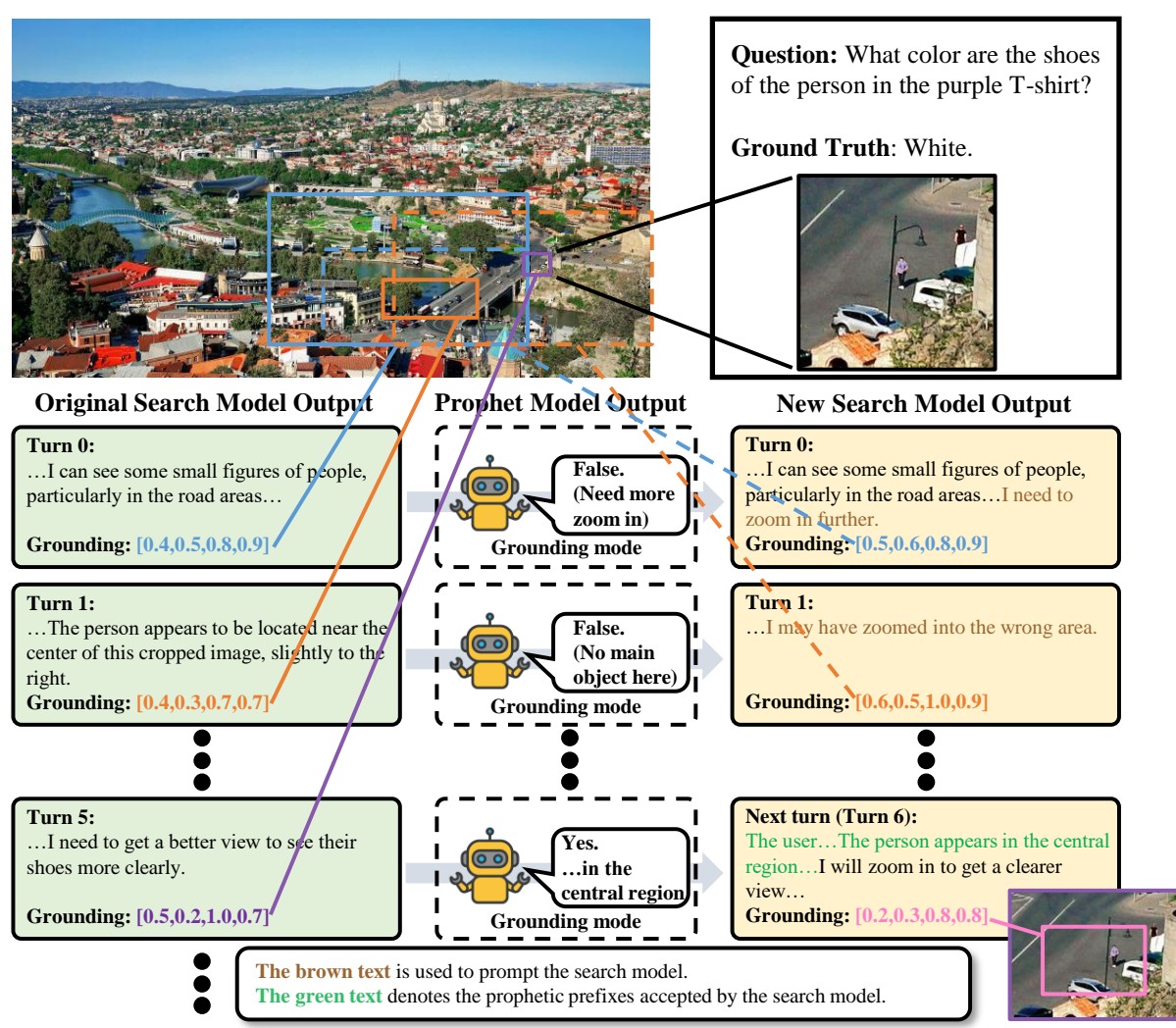

*Figure 4.* An example of SeProD. At turn 0, the region localized by the search model through its grounding capability is provided as input, and the prophet model determines that a further zoom-in operation is required, prompting the search model to zoom in more precisely. At turn 1, the prophet model judges that the region obtained by the search model does not contain the target of interest and instructs it to search for an alternative, correct region. At turn 5, the prophet model identifies the target in the image and outputs detailed information (i.e., the person appears in the central region of the image) as prophetic prefixes for the next turn (turn 6).

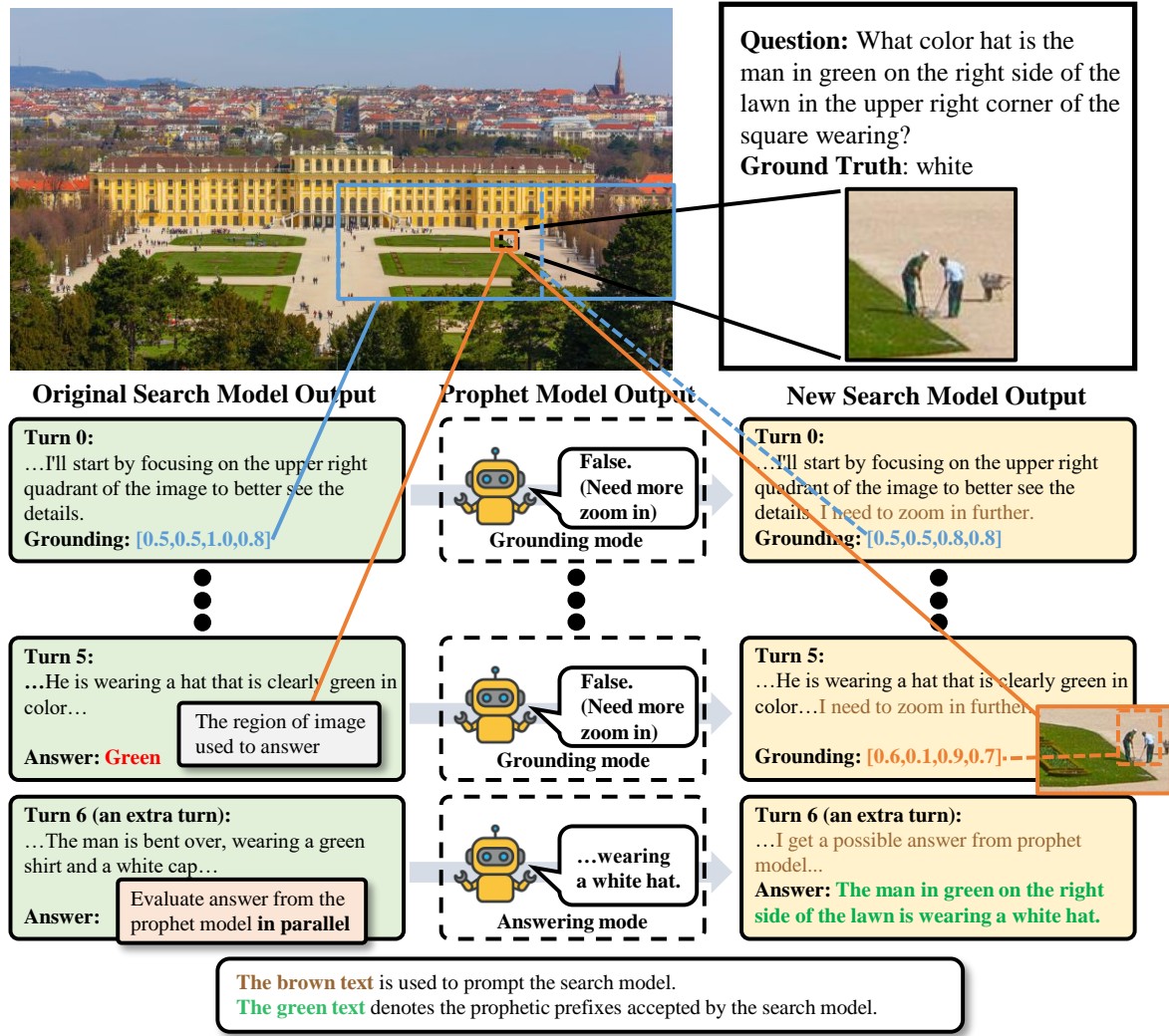

*Figure 5.* An example of SeProD. At turns 0 and 5, the regions localized by the search model through its grounding capability are provided as input, and the prophet model prompts the search model to perform further zoom-in operations. At turn 6, the prophet model takes the final image as input and generates the answer, which is used as prophetic prefixes. When the search model needs to generate the final answer, the prophetic prefixes are evaluated in parallel.

## B. Failure Cases of SeProD

In this section, we present representative failure cases of SeProD, which primarily arise from two sources: grounding errors in the search model and insufficient clarity in cropped image regions.

### B.1. Grounding Errors in the Search Model

As shown in Fig. 6, at the final turn (turn 8), the search model ultimately localizes a misleading and incorrect region for answering, which causes the prophet model to generate its response based only on this erroneous region.

### B.2. Insufficient Clarity in Cropped Image Regions

As shown in Fig. 7, at the final turn (turn 3), the search model localizes the correct region. However, the target text in the image is inherently unclear and ambiguous, which leads the prophet model to produce an incorrect answer.

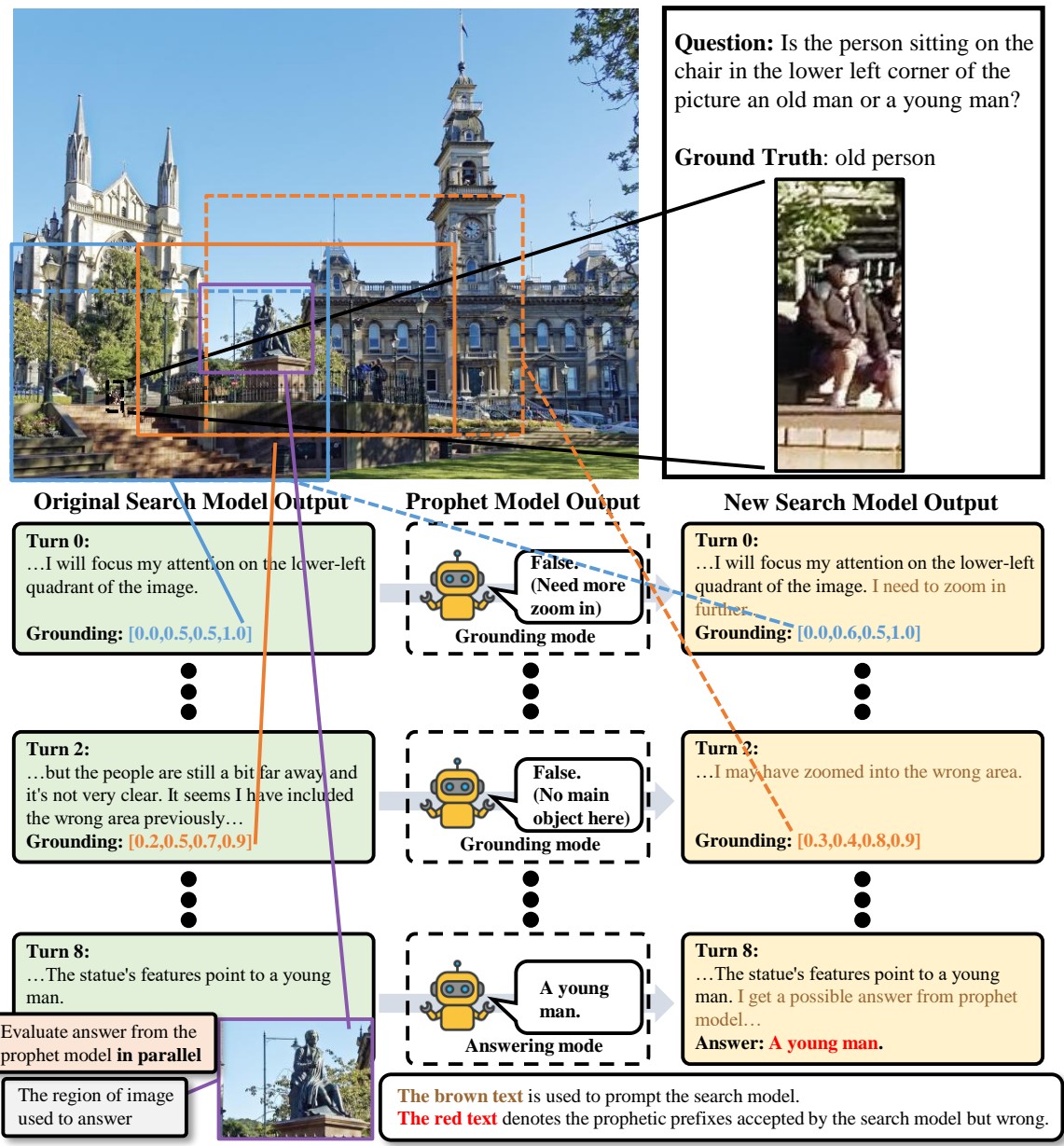

*Figure 6.* A failure case of SeProD. At the final turn (turn 8), the search model ultimately localizes a misleading and incorrect region for answering, which causes the prophet model to generate its response based only on this erroneous region.

## C. Failure of Naïve Textual Interfaces

In this section, we present a concrete example illustrating the failure cases of the naïve approach. Fig. 8 covers two scenarios: (1) failure to directly steer the outputs of the post-trained LVLM, and (2) disruption of contextual coherence, which destabilizes multi-step reasoning.

## D. Details on the Benchmarks Used in This Paper

### D.1. Benchmarks for Visual Search Tasks

**V\* Bench** (Wu & Xie, 2023) is a benchmark constructed on 191 high-resolution images sampled from the SA-1B dataset (Kirillov et al., 2023). For each image, a multiple-choice question is provided, where exactly one option is correct.

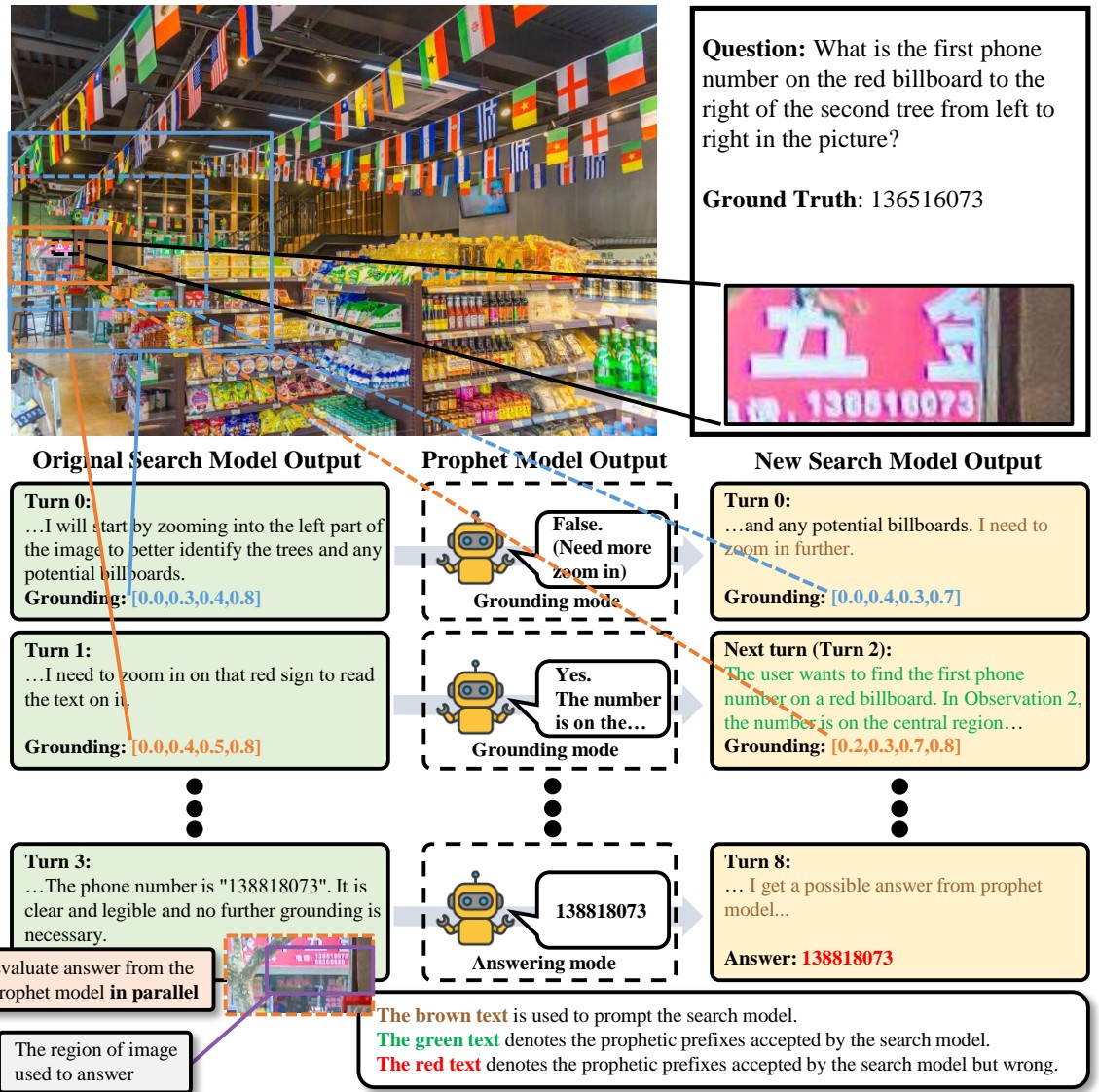

*Figure 7.* A failure case of SeProD. At the final turn (turn 3), the search model localizes the correct region. However, the target text in the image is inherently unclear and ambiguous, which leads the prophet model to produce an incorrect answer.

The benchmark comprises two task categories: attribute recognition and spatial relationship reasoning. The attribute recognition split includes 115 images and targets the identification of object-level properties. The spatial reasoning split contains 76 images and focuses on inferring relative spatial configurations between object pairs.

**HR-Bench** (Wang et al., 2024b) is bulit from 200 high-resolution images curated from the DIV8K dataset (Gu et al., 2019). The benchmark is evenly partitioned into two subsets: Fine-grained Single-instance Perception (FSP) and Fine-grained Cross-instance Perception (FCP), each containing 100 images. The FSP subset targets fine-grained understanding of individual objects, covering tasks such as attribute identification, visual prompting, and OCR recognition. In contrast, the FCP subset focuses on relational reasoning, including map interpretation, chart analysis, and spatial reasoning. HR-Bench is released in two configurations, namely HR-Bench 8K and HR-Bench 4K. The 8K version retains the original image resolution, whereas the 4K version is obtained by cropping regions centered on target objects. Each image is paired with a single multiple-choice question consisting of four candidate answers and one ground-truth option. The answer options are rotated across images, yielding 800 total samples.

**VisualProbe Test** (Lai et al., 2025) is divided into three difficulty levels, namely easy, medium, and hard, which contain

*Figure 8.* Failure of naïve textual interfaces. Pre-training LVLM's output weakly steers the post-training model and breaks coherence across steps, leading to unstable multi-step reasoning.

141, 268, and 106 samples, respectively. Compared to previous visual search benchmarks, VisualProbe places greater emphasis on small target objects and a large number of distracting elements. These characteristics require models to perform iterative exploration and rely on trial-and-error reasoning. Each sample consists of a single image paired with an open-ended question.

### D.2. Benchmarks for General VQA

**MME-RealWorld** (Zhang et al., 2025c) comprises 13,366 high-resolution images with an average resolution of 2,000 × 1,500 pixels, accompanied by 29,429 annotations spanning 43 sub-tasks.

**ScienceQA** (Lu et al., 2022) is a benchmark designed for multiple-choice science question answering. It spans three academic domains, including social science, natural science, and language science. The test split comprises 4,241 samples.

**OCRBench** (Liu et al., 2024) is designed to systematically assess the OCR recognition performance of models. The benchmark is organized into five task categories, covering text recognition, scene Text-Centric VQA, document-Oriented VQA, key information extraction, and handwritten mathematical expression recognition. In total, OCRBench contains 1,000 samples.

**CVBench** (Tong et al., 2024) is derived from several established vision benchmarks and comprises 2,638 examples. 2D visual understanding is evaluated through spatial relationships and object counting, while 3D visual understanding is assessed using depth order and relative distance.

## E. Implementation Details for Analysis in the Introduction

**Examining the intrinsic capabilities of post-training LVLMs.** This paragraph provides the implementation details corresponding to Fig. 2(a). We use the RefCOCO (Yu et al., 2016) validation set RefCOCO$_{val}$ to evaluate the grounding ability of pre-training and post-training LVLMs. The metric is IoU@0.5. We use the corresponding OCR recognition, spatial understanding, and counting questions from VisualProbe test (Lai et al., 2025) to evaluate the associated capabilities. Specifically, we select the current state-of-the-art Mini-o3 (Lai et al., 2025) as the post-training LVLM. We choose Qwen2.5-VL-7B (Bai et al., 2025b), which is used to initialize Mini-o3, as the pre-training model. For grounding evaluation of Mini-o3, we treat the cropped image used for its final answer as the predicted result. For the comparison of OCR recognition, spatial understanding, and counting abilities between the two models, we first obtain the answers and inference trajectories of Mini-o3 by applying the corresponding VisualProbe test questions. Subsequently, Qwen2.5-VL-7B is provided with the image corresponding to the fifth turn in the Mini-o3 inference trajectory, together with the original question, to obtain its prediction. If the inference trajectory contains fewer than five turns, the final answering turn of Mini-o3 is used instead.

**Testing interference in long multi-step reasoning contexts.** This paragraph provides the implementation details corre-

sponding to Fig. 2(b). We report the performance of the state-of-the-art Mini-o3 on three subsets of VisualProbe test before and after removing irrelevant context, as well as the corresponding improvements. Specifically, we define irrelevant context as the reasoning rounds in which the model explores erroneously magnified regions. For each sample, given the reasoning trajectory generated by Mini-o3, we remove the irrelevant context and the final answering round. We then feed the revised trajectory back into the model, prompting it to generate a new answering round and obtain a new prediction.

**Probability distribution curves.** This paragraph provides the implementation details corresponding to Fig. 2(c). We obtain the probability distribution curves for the original visual-search LVLM, the naïve method, and our SeProD. Specifically, we select Mini-o3 as the search model and Qwen2.5-VL-3B (Bai et al., 2025b) as the prophet model, and conduct experiments on the VisualProbe-test dataset. For each method, we compute the average probability of the final answer tokens under the mini-o3 and randomly sample 100 instances to visualize the resulting probability distributions.

