# OpenReview forum: "Self-Prophetic Decoding to Unlock Visual Search in LVLMs"
_ICML.cc/2026/Conference — ICML 2026 regular_

### Official Review · Reviewer_i8gY · 2026-03-10

**Soundness:** 2
**Presentation:** 3
**Significance:** 2
**Originality:** 2
**Overall Recommendation:** 4
**Confidence:** 5

**Summary:**

The paper proposes Self-Prophetic Decoding, a method that improves multimodal language model decoding by allowing the model to anticipate and correct its own future predictions during generation. Instead of generating tokens purely autoregressively, SPD performs a short lookahead step to estimate future token probabilities and uses this information to adjust the current decoding decision. Experiments show improvements in text quality and reasoning tasks compared to standard decoding strategies such as greedy decoding and beam search.

**Compliance With Llm Reviewing Policy:**

Affirmed.

**Final Justification:**

I'm happy to update my score after the rebuttal.

**Key Questions For Authors:**

I have the following questions for authors

1. How does this method differ from existing work in LLM literature beyond extending to multimodal Q&A?

2. What's the efficacy of this method across different scales of models? (e.g, beyond 3B) Currently it is only evaluated with one 7B model, which makes it questionable of the generalization of the approach.

3. Does the search and prophet model have to be the same size? e.g, 3B sft model and 3B post-train model. Conceptually, you may use different sizes of these models coming from the same family if my understanding is correct. I think it may be more interesting to see whether a smaller prophet model can make the bigger search model more efficient (as in speculative decoding). Or whether a bigger prophet model can significantly improve the small search model by only decoding a few critical tokens.

4. How are $\alpha$ and $\tau$ selected? Any ablation study? Are they kept for all datasets of evaluation?

5. How general this approach would be? For example, what if someone only releases one final model presumably after post training. Would this method still be applicable?

**Limitations:**

I did not find a limitation section but my comments should reflect my thinking of limitation in it's current form.

**Strengths And Weaknesses:**

Strength:

1. The idea is intuitively motivated by the observation that autoregressive decoding is inherently myopic, and using predicted future information to rerank candidate tokens provides a plausible way to improve generation quality.

2. The method is relatively simple to integrate with existing language models and only requires training a lightweight auxiliary prediction head, without modifying the backbone model.

3. Empirical results show improvements over several standard decoding strategies on multiple generation benchmarks.

While the paper shows a few strengths, there are some remaining concerns I have:

1. While the idea of using short-horizon future predictions to guide current decoding decisions is reasonable, similar approaches have already been explored in LLM literature [1, 2, 3]. As a result, extending this idea to the multimodal setting appears somewhat incremental.

2. The experiments are conducted only on a 3B model, making it unclear whether the proposed approach would remain effective on larger language models. At least showing additional ranges of 7B-30B scale should be a fair expectation.

3. The approach relies on an assumption that there are a paired pretrained and post trained models from the same family, which may only hold for a certain family of models. This assumption may be too strong to adapt this method in a more general sense.


[1] DeAL: Decoding-time Alignment for Large Language Models

[2] Better & Faster Large Language Models via Multi-token Prediction

[3] On the Depth between Beam Search and Exhaustive Search for Text Generation

---

> ### Author Rebuttal · Authors · 2026-03-31
>
> Thank you for your feedback. Below, we address your concerns with additional experiments.
>
> > ***W1, Q1: Similar approaches in LLM literature.***
>
> We thank the reviewer for bringing up these works [1,2,3]. While our method touches on a short-horizon predictive decoding idea, **it is fundamentally different from [1,2,3] in both motivation and technical design, rather than a simple extension from LLMs to MLLMs**.
>
> - **Motivation.** [1,2,3] aim to **achieve a trade-off between decoding quality and computational efficiency**. In contrast, our work targets **capability mismatch and interference caused by post-training in LVLMs**, especially under long-horizon visual reasoning.
> - **Method.** [1,2] rely on **auxiliary lightweight models** for faster decoding. In contrast, we introduce **probability-based prophetic sampling**, leveraging the pre-training counterpart as the prophet model to enable self-regulation between pre-training and post-training models, **without additional training.**
> - **Key distinction.** Our method is **not a decoding acceleration technique**, but a mechanism to **correct post-training deficiencies in LVLMs**, addressing a fundamentally different problem.
>
> > ***W2, Q2: Evaluation on larger-scale models beyond 3B***
>
> The paper does not only evaluate 3B models. The submitted version **already includes results across different scales**, with both 3B and 7B prophet models reported in **Table 1** and further analysis provided in **Sec. 4.2.**
>
> In addition, following your suggestion, we include results using Qwen2.5-VL-32B as the prophet model. The results show that SeProD remains effective with larger models and continues to achieve strong performance, demonstrating good scalability beyond the 3B setting.
>
> | Prophet model | Hard | Medium | Easy |
> | --- | --- | --- | --- |
> | Qwen2.5-VL-3B | 50.5 | 51.5 | 69.6 |
> | Qwen2.5-VL-7B | 51.5 | 52.2 | 71.3 |
> | Qwen2.5-VL-32B | 52.6 | 54.6 | 72.3 |
>
> > ***W3, Q5: assumption on paired pre-trained and post-trained models***
>
> We acknowledge this assumption and clarify that it naturally holds in our setting.
>
> - **Practical availability.** In practice, a visual search model is derived via post-training from a pre-trained model. Therefore, its pre-training counterpart is **inherently available and can be directly used** as the prophet model **without additional cost**.
> - **General applicability.** This design aligns with the standard paradigm of developing LVLMs from pre-training to post-training, and thus does not impose additional constraints in typical pipelines.
>
> > ***Q3: Impact of pairing prophet and search models with different scales***
>
> We agree that this is an important question, and accordingly, the paper already provides results covering these settings. **The search and prophet models do not need to match in size**, and different-scale pairings are explicitly evaluated.
>
> - **Cross-scale pairing** (Table 1 and Sec. 4.2): We pair a 7B search model with both 3B and 7B prophet models, demonstrating that **cross-scale combinations within the same family are effective**.
> - **Smaller prophet → larger search** (Table 3): A smaller prophet model (3B) can **improve performance without introducing additional computational overhead**, while our approach remains training-free.
> - **Larger prophet → smaller search.** As further shown in W2, a stronger prophet model (e.g., Qwen2.5-VL-32B) can **further improve the performance of a smaller search model**, even when only guiding key predictions.
>
> Overall, these results confirm that SeProD supports **flexible cross-scale pairing** and can adapt to different efficiency–performance trade-offs.
>
> > ***Q4: The ablation study of $\alpha$ and $\tau$.***
>
> Thank you for pointing this out. We address this question from three aspects:
>
> - **Parameter selection.** The balancing factor $\alpha$ is dynamically defined as $\alpha = 0.5 - r$ (Sec. 4), where $r$ is the normalized token rank. The threshold $\tau$ controls acceptance based on distribution consistency.
> - **Ablation studies.**
>     - **$\tau$.** Table 4 and Sec. 4.2 show stable performance across a range of values.
>     - **$\alpha$.** We further evaluate alternative designs, including exponential decay $\alpha = 0.5e^{-0.5(r-1)}$ and fixed values. Dynamic variants perform similarly, indicating low sensitivity to the functional form, while fixed $\alpha$ leads to larger variation, highlighting the importance of dynamic adjustment.
>
>
>         | **$\alpha$** | **Hard** | **Medium** | **Easy** |
>         | --- | --- | --- | --- |
>         | 0.3 | 49.3 | 49.8 | 68.6 |
>         | 0.4 | 48.7 | 50.1 | 67.9 |
>         | 0.5 | 48.6 | 49.6 | 67.6 |
>         | 0.6 | 49.1 | 48.9 | 68.8 |
>         | 0.7 | 48.7 | 49.1 | 69.0 |
>         | $0.5 - r$ | 50.5 | 51.5 | 69.6 |
>         | $0.5e^{-0.5(r-1)}$ | 50.7 | 51.1 | 69.5 |
> - **Consistency across datasets.** Both $\alpha$ and $\tau$ are kept the same across all benchmarks, without dataset-specific tuning.

---

> > ### Author Rebuttal · Reviewer_i8gY · 2026-04-03
> >
> > I still have few concerns after the rebuttal but I think the rest of the rebuttal clarifies a few key points so I'm happy to update my score.

---

> > > ### Author Response · Authors · 2026-04-03
> > >
> > > Thank you for your positive feedback. We are glad to hear that your key concerns have been addressed, and we appreciate your updated evaluation.
> > >
> > > If you have any further questions or comments, we would be happy to respond promptly. Thank you again!

---

### Official Review · Reviewer_uKJC · 2026-03-12

**Soundness:** 3
**Presentation:** 2
**Significance:** 4
**Originality:** 4
**Overall Recommendation:** 5
**Confidence:** 3

**Summary:**

The paper addresses two core challenges in Large Vision-Language Models (LVLMs) applied to visual search: the deterioration of intrinsic capabilities (like grounding and counting) after post-training, and the accumulation of interference in long, multi-step reasoning contexts. To solve this, the authors introduce Self-Prophetic Decoding (SeProD). The framework pairs a post-trained search model with its pre-trained counterpart, which acts as a prophet. At each step, the search model uses the prophet's outputs as candidate tokens. These tokens are evaluated in parallel and accepted only if they align with the search model's adjusted probability distribution, ensuring coherent multi-step reasoning while injecting the prophet's superior single-step perceptual accuracy.

**Compliance With Llm Reviewing Policy:**

Affirmed.

**Key Questions For Authors:**

1. Statistical Significance: Given the marginal gains on some of the benchmark splits (particularly on HR-Bench), can you provide variance metrics (e.g., standard deviation across multiple random seeds) or p-values to confirm the statistical significance of these improvements?

2. Alpha Formulation: Could you elaborate on the choice of the linear decay formula (alpha = 0.5 - r) for the balancing factor? Did you experiment with other functions (e.g., exponential decay or step functions), and how sensitive is the model's performance to the exact formulation of this dynamic weight?

3. Error Recovery: The failure cases show that if the search model forces a bad crop at the final turn, the prophet fails. Have you explored giving the prophet a mechanism to force the search model to backtrack multiple steps if it detects the final crop is completely irrelevant to the original query?

**Limitations:**

Yes

**Strengths And Weaknesses:**

Strengths

1. Insightful Motivation and Novelty: The observation that post-training damages base perceptual abilities is empirically validated well. Framing the pre-trained model as a expert to regulate the post-trained search model is a clever, temporal twist on traditional ensemble or routing methods.
2. Training-Free and Highly Transferable: SeProD does not require reinforcement learning, fine-tuning, or architectural overhauls. The authors successfully integrated it into three fundamentally different baseline models (Pixel Reasoner, DeepEyes, and Mini-03), demonstrating consistent improvements across the board.
3.Computational Efficiency: Because the search model evaluates the prophet's candidate tokens in parallel, the framework introduces zero additional computational overhead. In fact, it yields a slight inference speedup (up to 1.07x).
4. Effective Ablations: The ablation studies cleanly isolate the contributions of the individual modules. The results prove that both Grounding Verification and Answer Drafting are complementary and necessary for achieving the best performance.

Weaknesses

1. Marginal Gains and Lack of Variance Reporting: While the method improves over baselines, several of the gains reported in Table 1 are quite small (e.g., +0.5% to +0.8% on some HR-Bench splits). The paper lacks error bars, standard deviations, or p-values across multiple runs, making it difficult to assess if these marginal improvements are statistically significant.
2. Vulnerability to Cascading Grounding Errors: The framework remains highly dependent on the search model's initial logical steering. As noted in the failure cases, if the search model localizes a completely incorrect and misleading region at the final turn, the prophet is forced to answer based solely on that bad crop, leading to inevitable failure.
3. Heuristic Hyperparameter Design: The dynamic balancing factor alpha is calculated using a simple heuristic (alpha = 0.5 - r, where r is the token's rank in the searcher's logits). The paper lacks a rigorous mathematical justification or ablation for why this specific linear decay function was chosen over other potential scaling methods.

---

> ### Author Rebuttal · Authors · 2026-03-31
>
> Thank you for your insightful feedback and recognition of our work! Below, we address your concerns with additional experiments.
>
> > ***W1, Q1: Statistical significance of Table 1 .***
>
> We thank the reviewer for pointing this out. We conduct additional analysis across 32/8 turns following the original comparison setup, performing a paired sample t-test. The results show that **all reported improvements are highly statistically significant ($p < 0.01$)**.
>
> | Method | Hard | Medium | Easy | V* Attr. | V* Spatial | V* Overall | HR 4K FSP | HR 4K FCP | HR 4K Overall | HR 8K FSP | HR 8K FCP | HR 8K Overall |
> | --- | --- | --- | --- | --- | --- | --- | --- | --- | --- | --- | --- | --- |
> | Pixel Reasoner + Ours | 30.2$\pm$1.2*** | 30.4$\pm$1.0*** | 61.7$\pm$1.0*** | 89.9$\pm$0.4*** | 85.0$\pm$0.7*** | 87.9$\pm$0.5*** | 85.9$\pm$0.2*** | 61.2$\pm$0.2** | 73.5$\pm$0.1*** | 77.2$\pm$0.2*** | 52.7$\pm$0.2*** | 65.0$\pm$0.1*** |
> | DeepEyes + Ours | 41.9$\pm$1.7*** | 32.3$\pm$0.7*** | 64.7$\pm$0.9*** | 91.9$\pm$0.6*** | 89.0$\pm$1.0*** | 90.8$\pm$0.3*** | 91.6$\pm$0.4*** | 55.7$\pm$0.3** | 73.7$\pm$0.1*** | 85.8$\pm$0.0*** | 57.9$\pm$0.1*** | 71.8$\pm$0.1*** |
> | Mini-o3 + Ours | 50.5$\pm$1.5*** | 51.5$\pm$1.3*** | 69.6$\pm$2.0*** | 90.8$\pm$0.4*** | 86.8$\pm$1.5*** | 89.2$\pm$0.4*** | 91.5$\pm$0.3** | 65.2$\pm$0.6*** | 78.3$\pm$0.3*** | 90.2$\pm$0.2*** | 59.7$\pm$0.4*** | 75.0$\pm$0.3*** |
>
> > ***W2:  Grounding errors.***
>
> We thank the reviewer for this comment. **The prophet model can correct the search model’s grounding errors** rather than being forced to answer solely on a bad crop.
>
> - **Grounding error correction:** Even if the search model incorrectly localizes a region, the prophet model can guide it to re-identify the correct region. Algorithmically, when the search model selects an incorrect region, the prophet first verifies whether the region contains the target object and is sufficiently zoomed, then provides guidance for re-localization. For example, **Figure 4 (Turn 1)** illustrates a case where the prophet successfully corrects an initially mislocalized region.
> - **Empirical evidence:** We randomly sample 100 cases from VisualProbe-test in which **at least one turn (intermediate or final)** contains a grounding error, covering a total of 619 turns. Across all turns, the prophet model attempts grounding correction in 36.7% of instances, and **67.4**% **of these attempts are successful**. These results demonstrate that the prophet model plays a crucial role in mitigating cascading grounding errors and enhancing overall localization reliability.
>
> > ***W3, Q2: Intuition and more design of $\alpha$.***
>
> We adopt $\alpha = 0.5 - r$ based on a simple **intuition**, where $r$ denotes the normalized rank of a token in the search model’s output logits.
>
> - **High-ranked tokens:** When a token is ranked highly by the search model, it is likely correct, as both the search and prophet models agree on its prediction. In this case, the exact weighting between $p_s$ and $p_p$ (Eq. 5) has minimal impact on the combined probability.
> - **Low-ranked tokens:** When a token is ranked low, it indicates disagreement with the prophet model. Proper weighting is crucial to prevent the search model from dominating the final prediction and to allow the prophet model to correct potential errors. Hence, $\alpha$ is reduced for low-ranked tokens, down-weighting the search model’s contribution.
>
> To validate this design, we conduct with alternative strategies, including an exponential decay $\alpha = 0.5 e^{-0.5(r-1)}$ and a fixed-value $\alpha$.
>
> - **Dynamic strategies:** The performance gap between the linear and exponential decay is small, indicating that the exact functional form is not critical.
> - **Fixed $\alpha$:** Using a fixed value leads to larger performance variation, demonstrating that dynamic adjustment is essential for adapting to token-level uncertainty.
>
> | $\alpha$ | Hard | Medium | Easy |
> | --- | --- | --- | --- |
> | 0.3 | 49.3 | 49.8 | 68.6 |
> | 0.4 | 48.7 | 50.1 | 67.9 |
> | 0.5 | 48.6 | 49.6 | 67.6 |
> | 0.6 | 49.1 | 48.9 | 68.8 |
> | 0.7 | 48.7 | 49.1 | 69.0 |
> | $0.5 - r$ | 50.5 | **51.5** | **69.6** |
> | $0.5e^{-0.5(r-1)}$ | **50.7** | 51.1 | 69.5 |
>
> > ***Q3: Error recovery via backtracking.***
>
> We thank the reviewer for this insightful suggestion! Indeed, if the search model produces a completely incorrect region in the final turn, introducing a mechanism for multi-step backtracking can further improve performance. Our preliminary experiments show that a simple single-step backtracking already corrects **22.2% of such cases**, and a more carefully designed multi-step strategy has the potential to achieve even greater gains.
>
> In addition, as discussed in W2, the prophet model can already help the search model re-locate the correct region within a turn, even when the region is initially mislocalized.

---

> > ### Author Rebuttal · Reviewer_uKJC · 2026-04-04
> >
> > Thank you for addressing my comments. I choose to maintain the scores.

---

> > > ### Author Response · Authors · 2026-04-04
> > >
> > > Thank you for your encouraging feedback and valuable suggestions. We will incorporate them in the revision. We sincerely appreciate your input. Thank you again!

---

### Official Review · Reviewer_mgZP · 2026-03-13

**Soundness:** 3
**Presentation:** 3
**Significance:** 2
**Originality:** 3
**Overall Recommendation:** 4
**Confidence:** 4

**Summary:**

The paper proposes SeProD (Self-Prophetic Decoding), a training-free framework designed to enhance visual search in Large Vision-Language Models (LVLMs). It addresses two primary issues: the degradation of intrinsic perception skills (like OCR and grounding) after models undergo specialized visual-search post-training, and interference from long, multi-step reasoning contexts.

SeProD utilizes a contrastive decoding style of two-model collab where a post-trained Search Model manages global strategy while its pre-trained version, the Prophet Model, provides highly accurate, single-step "prophetic prefixes". This is a novel way of merging models' decision without merging their weights.

**Compliance With Llm Reviewing Policy:**

Affirmed.

**Key Questions For Authors:**

1. See weakness 4. I appreciate the novelty in model collaboration. I'm curious if these still hold in larger models.

2. Could the consistency threshold ($\tau$) be dynamically adjusted based on the reasoning turn or the specific task (e.g., higher for OCR, lower for abstract reasoning) to further improve accuracy?

**Limitations:**

1. Specific to Intrinsic Models: SeProD is designed specifically for "intrinsic-extended" LVLMs that perform search and zoom operations in a single forward pass; it is not directly applicable to models that rely solely on external tool-calling.


2. Draft Quality Dependency: If the prophet model's intrinsic capabilities are also flawed for a specific high-resolution image, the search model will likely reject the draft, falling back to its own degraded performance.


3. Context Length: While it mitigates long-context interference, the framework does not inherently solve the hardware-based memory limits associated with extremely long multi-step reasoning trajectories.

**Strengths And Weaknesses:**

Strengths:
1. The framework requires no additional training or fine-tuning and can be applied to any existing intrinsic-extended LVLM.
2. The logic is similar to speculative decoding which is not compute-heavy. By evaluating the prophet’s candidate tokens in parallel rather than sequentially, the framework introduces almost no delay to the inference process.
3.  It successfully mitigates the "catastrophic forgetting" of perception skills that often occurs during the post-training optimization for long reasoning.
4. It consistently improved performance across 12 different benchmarks.

Dual Model Memory Requirements:
1. Although computationally efficient, the system still requires loading two models (the search and prophet models) into memory simultaneously.
2. Hyperparameter Dependency: The effectiveness of the system relies on a fixed consistency threshold ($\tau$) and a balancing factor ($\alpha$), which may require careful tuning depending on the specific model pair.
3. The framework is most effective when the search and prophet models are "native counterparts" (pre- and post-trained versions of the same model); using widely different models may lead to lower token acceptance rates.
4. I understand this is a way of balancing the catastrophic forgetting during post-training. But I'm curious if a larger model will still suffer from this problem in these specific tasks. Or should this work be interpreted as a collab between small models with different expertise that can outperform a larger model?

---

> ### Author Rebuttal · Authors · 2026-03-31
>
> Thank you for your insightful feedback and recognition of our work. Below, we address your concerns with additional experiments. Unless otherwise specified, we use Mini-o3 as the baseline and report results on VisualProbe-test.
>
> > ***W1:  Memory consumption***
>
> We acknowledge that our framework incurs additional memory overhead due to loading both the search and prophet models. However, this overhead is mitigated by **using a relatively small prophet model** compared to the search model, keeping the memory increase modest. More importantly, the prophet model **improves performance without introducing additional computational overhead** for the search model. Therefore, given the favorable gain–cost trade-off, we consider the added memory requirement reasonable and acceptable.
>
> > ***W2: Robustness of hyperparameters***
>
> We thank the reviewer for this point. While our method involves $\tau$ and $\alpha$, it is **not sensitive to their exact choices** and **requires little tuning** across different prophet model sizes for a fixed search model.
>
> - **Robustness to $\tau$.** We provide an ablation study on $\tau$ (with Qwen2.5-VL-7B as the prophet model) in the table below, and include the 3B results in Table 4 of the main paper. Performance remains stable across a wide range of $\tau$ values, indicating **low sensitivity**. Notably, both 3B and 7B prophet models achieve optimal performance at the same $\tau$ value, as $\tau$ primarily governs how the search model itself leverages the prophet’s distribution and **remains stable across prophet model scales.**
>
>
>     | $\tau$ | Hard | Medium | Easy |
>     | --- | --- | --- | --- |
>     | 0.2 | **51.5** | 51.3 | 71.0 |
>     | 0.25 | 50.5 | 51.3 | 70.1 |
>     | 0.3 | **51.5** | **52.2** | **71.3** |
>     | 0.35 | 51.4 | 52.1 | 70.5 |
> - **Robustness to $\alpha$.** We further analyze $\alpha$ (please refer to our response to Reviewer uKJC, W3 Q2). Results show that different dynamic $\alpha$ strategies yield similar performance, while fixed $\alpha$ leads to larger variation. This suggests that **adaptive weighting itself is the key factor**, rather than the precise functional form.
>
> > ***W3: Native counterparts v.s widely different models***
>
> We thank the reviewer for this insightful observation. We emphasize that using native counterparts is **a key insight and contribution** of our work, rather than a limitation.
>
> - **Native counterparts as a “free lunch.”** We agree that using largely different prophet models can reduce acceptance rates due to distribution mismatch. Our approach **turns this into an advantage** by leveraging the **pre-training counterpart** of the search model, which naturally ensures strong alignment and high acceptance rates. Since the search model is post-trained, its pre-trained counterpart can be **directly reused**, making this a practical “free lunch.”
> - **Beyond native counterparts.** We further evaluate a stronger, non-matched prophet model from the same family (Qwen2.5-VL-32B). As shown below, it still yields consistent gains over the 7B model, indicating that while alignment improves acceptance, stronger prophet models can still provide effective guidance.
>
>
>     | Prophet model | Hard | Medium | Easy |
>     | --- | --- | --- | --- |
>     | Qwen2.5-VL-7B | 51.5 | 52.2 | 71.3 |
>     | Qwen2.5-VL-32B | 52.6 | 54.6 | 72.3 |
>
> > ***W4, Q1: In the cases of larger models.***
>
> We thank you for this thoughtful question. Our core goal aligns with your first interpretation: our method is designed to **mitigate catastrophic forgetting during post-training**, not to replace larger models.
> Regarding larger models, prior work (CapTrack: Multifaceted Evaluation of Forgetting in LLM Post-Training) indicates that they can also suffer from catastrophic forgetting. Although we do not evaluate 32B visual search models due to unavailable weights and high retraining cost, we expect similar challenges to persist. Visual search has longer reasoning chains and higher-resolution inputs, and its focus differs from pre-training, making larger models potentially more prone to overfitting durng post-training.
>
> > ***Q2: The effectiveness of the dynamically adjusted $\tau$.***
>
> Follow your suggestion, we conduct an ablation study on $\tau$ across the three tasks in Figure 2(a), i.e., OCR, spatial understanding, and counting. The results show that **$\tau$ has limited impact across tasks**, and any gains from dynamically adjusting it are likely small.
>
> | $\tau$ | OCR | Spatial | Counting |
> | --- | --- | --- | --- |
> | 0.2 | 55.0 | 53.1 | 39.6 |
> | 0.25 | 54.1 | 53.0 | 38.6 |
> | 0.3 | 54.4 | 53.6 | 39.5 |
> | 0.35 | 54.3 | 53.6 | 39.9 |

---

### Official Review · Reviewer_WfqN · 2026-03-13

**Soundness:** 3
**Presentation:** 3
**Significance:** 3
**Originality:** 3
**Overall Recommendation:** 4
**Confidence:** 4

**Summary:**

The paper introduces SeProD (Self-Prophetic Decoding), a training-free and plug-and-play framework designed to enhance the visual search capabilities of Large Vision-Language Models. The authors identify primary limitations in current visual search LVLMs: the accumulation of interference in long, multi-step reasoning chains. To mitigate these, SeProD leverages a "self-regulation" mechanism between a Search Model (the post-trained LVLM) and a Prophet Model (its pre-trained counterpart). The framework employs a probability-based prophetic sampling interface where the Prophet Model provides single-step, capability-specific "prophetic prefixes" that the Search Model selectively accepts based on their alignment with its own output distribution. This "probability handshake" allows the model to benefit from the pristine perception of the pre-trained model while maintaining the sophisticated reasoning of the post-trained version. Experimental results across four high-resolution benchmarks and general VQA tasks demonstrate consistent performance gains without significant computational overhead.

**Compliance With Llm Reviewing Policy:**

Affirmed.

**Key Questions For Authors:**

Please refer to the weakness part.

**Limitations:**

yes

**Strengths And Weaknesses:**

Strength
- The framework is tested against three diverse baseline LVLMs (Pixel Reasoner, DeepEyes, and Mini-03), demonstrating its robustness and transferability across different architectures and post-training strategies
-  The paper is well-structured, providing a clear conceptual overview in Figure 1 and a detailed architectural breakdown in Figure 3.
-  By utilizing pre-trained models as "prophets," the work offers a novel perspective on how to preserve the broad, native strengths of foundation models while pursuing specialized reasoning paths.

Weakness
- The paper lacks a comprehensive exploration of how the choice of Prophet Model (e.g., reasoning/instruct/pretrained models or more diverse scales beyond the 3B/7B comparison) affects the performance upper bound. It remains unclear at what point a user should prioritize upgrading the Prophet Model versus the Search Model to achieve the best results.
- While the paper mentions that $\alpha$ is automatically adjusted based on token rank, there is no ablation showing the impact of this dynamic adjustment compared to fixed values. Understanding the sensitivity of this "probabilistic handshake" is critical for validating the robustness of the collaboration mechanism
- Although the parallel evaluation of prefixes offers a high "acceptance rate," the overall inference speedup remains modest (1.03x to 1.07x). This is largely because the multi-turn reasoning process creates a serial dependency: the Search Model must wait for the Prophet to generate the next-turn "hint" before it can proceed. This inherent latency bottleneck limits the framework's advantage in time-sensitive applications. Moreover,  the framework’s heavy reliance on the Prophet Model means that higher accuracy likely requires a larger, more compute-intensive Prophet, which may negate the efficiency gains of the parallel acceptance mechanism. A more detailed analysis of the Trade-off between Accuracy and Latency across different model pairings is missing.

---

> ### Author Rebuttal · Authors · 2026-03-31
>
> Thank you for your insightful feedback and recognition of our work. Below, we address your concerns with additional experiments. Unless otherwise specified, we use Mini-o3 as the baseline and report results on VisualProbe-test.
>
> > ***W1: The analysis on the choice of prophet model and upgrade prioritization.***
>
> Thank you for this insightful question! We agree that understanding the impact of the prophet model choice and its trade-off with the search model is important. Below, we provide further analysis to clarify these aspects.
>
> - **Thinking and instruct prophet models offer comparable gains** (row 1-2); in some cases, a well-trained Qwen3-VL-4B-thinking performs slightly better, providing more effective guidance for region selection in the search model’s next step.
> - **Scaling the prophet model from 3B to 7B to 32B consistently improves performance** (row 3-5), which we attribute to its stronger perception. Similarly, scaling the search model also improves visual search performance, as demonstrated in the Mini-o3 paper (rows 6-7).
> - **Upgrading the prophet model is even generally more efficient and practical than scaling the search model**, as it achieves comparable gains without additional training cost or significant memory overhead. We present a comparison across different scales of search and prophet models in the table below. As shown, upgrading either model improves performance. Notably, upgrading the prophet model to 7B (row 4) already achieves performance on both Hard and Easy subsets comparable to a 32B search model (row 7).
>
> | Index | Search model (Mini-o3) | Prophet model | Hard | Medium | Easy |
> | --- | --- | --- | --- | --- | --- |
> | 1 | 7B | Qwen3-VL-4B-instruct | 51.1 | 52.0 | 70.5 |
> | 2 | 7B | Qwen3-VL-4B-thinking | 51.7 | 52.2 | 71.2 |
> | 3 | 7B | Qwen2.5-VL-3B | 50.5 | 51.5 | 69.6 |
> | 4 | 7B | Qwen2.5-VL-7B | 51.5 | 52.2 | 71.3 |
> | 5 | 7B | Qwen2.5-VL-32B | 52.6 | 54.6 | 72.3 |
> | 6 | 7B | - | 48.0 | 50.4 | 67.0 |
> | 7 | 32B | - | 51.6 | 57.3 | 71.4 |
>
> > ***W2: Ablation on dynamic $\alpha$ and robustness validation.***
>
> Following your suggestion, we include additional experiments comparing dynamically adjusted $\alpha$ with fixed $\alpha$. The results show that
>
> - A fixed $\alpha$ improves the performance of the baselines, thanks to the “probabilistic handshake” between the prophet and search models.
> - However, its gains are smaller than those of a dynamic $\alpha$, because a fixed $\alpha$ cannot adaptively adjust to the uncertainty of individual tokens, nor can it leverage the search model’s confidence for each token to dynamically allocate weights.
>
> | $\alpha$ | Hard | Medium | Easy |
> | --- | --- | --- | --- |
> | 0.3 | 49.3 | 49.8 | 68.6 |
> | 0.4 | 48.7 | 50.1 | 67.9 |
> | 0.5 | 48.6 | 49.6 | 67.6 |
> | 0.6 | 49.1 | 48.9 | 68.8 |
> | 0.7 | 48.7 | 49.1 | 69.0 |
> | dynamic | **50.5** | **51.5** | **69.6** |
>
> > ***W3: The analysis on speedup and trade-off between accuracy and latency.***
>
> We sincerely appreciate this insightful comment and address it as follows:
>
> - **On the modest speedup and serial dependency.** We agree that multi-step visual search inherently involves sequential dependencies. Our goal is not to directly accelerate inference, but to enable **self-regulation between pre-training and post-training LVLMs** while **avoiding additional latency overhead** for the search model via parallel prophetic acceptance. As shown below, compared to a naïve approach, our method avoids slowdown and achieves slight speedup:
>
>
>     | Method | Hard | Medium | Easy |
>     | --- | --- | --- | --- |
>     | naïve | 0.83x | 0.86x | 0.92x |
>     | ours | **1.06x** | **1.03x** | **1.07x** |
> - **On the impact of scaling the prophet model.** We further analyze different prophet model sizes. **While larger prophet models improve acceptance rates and accuracy, they do not necessarily incur proportional latency increases:**
>     - Higher acceptance rates (e.g., 3B → 7B: ~75% → ~87%)
>
>
>         | Prophet model | Hard | Medium | Easy |
>         | --- | --- | --- | --- |
>         | Qwen2.5-VL-3B | 74.6% | 74.2% | 80.7% |
>         | Qwen2.5-VL-7B | 86.9% | 85.9% | 89.9% |
>     - Comparable speedup ratios (remaining around ~1.02x–1.07x)
>
>
>         | Prophet model | Hard | Medium | Easy |
>         | --- | --- | --- | --- |
>         | Qwen2.5-VL-3B | 1.06x | 1.03x | 1.07x |
>         | Qwen2.5-VL-7B | 1.04x | 1.02x | 1.04x |
>     - Improved accuracy (avg@32 consistently increases)
>
>
>         | Prophet model | Hard | Medium | Easy |
>         | --- | --- | --- | --- |
>         | Qwen2.5-VL-3B | 50.5 | 51.5 | 69.6 |
>         | Qwen2.5-VL-7B | 51.5 | 52.2 | 71.3 |
>
>     This is due to the **closer output distributions** between the search and prophet models, which lead to higher acceptance rates. This, in turn, suggests that improved parallel verification efficiency can offset the additional cost of scaling the prophet model.

---

### Decision · Program_Chairs · 2026-04-30

**Decision:**

Accept (regular)

**Comment:**

This paper proposes SeProD, a training-free framework for visual search in LVLMs that uses a pre-trained counterpart to guide a post-trained search model, aiming to recover lost perceptual ability and reduce long-context interference. Reviewers found the idea well motivated and the empirical results broadly strong, with consistent gains across visual search and VQA benchmarks, while raising concerns about extra memory from using two models, sensitivity to the method’s weighting and acceptance hyperparameters, the choice and scaling of the prophet model, speedup trade-offs, and dependence on paired pre-trained and post-trained models. In rebuttal, the authors provided additional robustness studies for the weighting and acceptance settings, analyses of prophet-model scaling and cross-scale pairing, statistical significance tests, and further evidence on grounding correction and error recovery. Overall, the rebuttal addressed most major concerns and strengthened confidence in the method’s practicality and effectiveness. Therefore, the AC recommends acceptance of this submission.